# Hemodynamic Response to Three Types of Urban Spaces before and after Lockdown during the COVID-19 Pandemic

**DOI:** 10.3390/ijerph18116118

**Published:** 2021-06-06

**Authors:** Agnieszka Olszewska-Guizzo, Ayako Mukoyama, Sho Naganawa, Ippeita Dan, Syeda Fabeha Husain, Cyrus S. Ho, Roger Ho

**Affiliations:** 1Institute for Health Innovation & Technology (iHealthtech) MD6, 14 Medical Drive, #14-01, Singapore 117599, Singapore; pcmrhcm@nus.edu.sg; 2NeuroLandscape Foundation, Suwalska 8/78, 03-252 Warsaw, Poland; 3Applied Cognitive Neuroscience Laboratory, Chuo University, 1-13-27 Kasuga, Bunkyo-ku, Tokyo 112-8551, Japan; a15.whyr@g.chuo-u.ac.jp (A.M.); a15.tg3y@g.chuo-u.ac.jp (S.N.); dan@brain-lab.jp (I.D.); 4Department of Psychological Medicine, Yong Loo Lin School of Medicine, National University of Singapore, NUHS Tower Block, Level 9, 1E Kent Ridge Road, Singapore 119228, Singapore; fabeha@nus.edu.sg (S.F.H.); su_hui_ho@nuhs.edu.sg (C.S.H.)

**Keywords:** fNIRS, brain, lockdown, park, urban, environment, COVID-19

## Abstract

(1) Background: Prolonged lockdowns with stay-at-home orders have been introduced in many countries since the outbreak of the COVID-19 pandemic. They have caused a drastic change in the everyday lives of people living in urbanized areas, and are considered to contribute to a modified perception of the public space. As research related to the impact of COVID-19 restrictions on mental health and well-being emerges, the associated longitudinal changes of brain hemodynamics in healthy adults remain largely unknown. (2) Methods: this study examined the hemodynamic activation patterns of the prefrontal and occipital cortices of 12 participants (5 male, M_age_ = 47.80, SD_age_ = 17.79, range 25 to 74, and 7 female, M_age_ = 39.00, SD_age_ = 18.18, range 21 to 65) passively viewing videos from three urban sites in Singapore (Urban Park, Neighborhood Landscape and City Center) at two different time points—T1, before the COVID-19 pandemic and T2, soon after the lockdown was over. (3) Results: We observed a significant and marginally significant decrease in average oxyhemoglobin (Oxy-Hb) over time for each of the visual conditions. For both green spaces (Urban Park and Neighborhood Landscape), the decrease was in the visual cortex, while for the City Center with no green elements, the marginal decrease was observed in the visual cortex and the frontal eye fields. (4) Conclusions: The results suggest that the COVID-19-related lockdown experienced by urban inhabitants may have contributed to decreased brain hemodynamics, which are further related to a heightened risk of mental health disorders, such as depression or a decline in cognitive functions. Moreover, the busy City Center scenes induced a hemodynamic pattern associated with stress and anxiety, while urban green spaces did not cause such an effect. Urban green scenes can be an important factor to offset the negative neuropsychological impact of busy urban environments post-pandemic.

## 1. Introduction

Since the beginning of the COVID-19 pandemic in early 2020, we have observed ever-increasing research highlighting its negative impact on the mental health and well-being of people worldwide. This is often attributed to economic uncertainty, fear of contracting the virus, losing a loved one, social isolation, or other drastic changes in everyday lifestyle during the prolonged stay-at-home orders introduced in many countries [1,2,3].

Less is known about the environmental aspects of stay-at-home orders and the longitudinal changes in brain functionality caused by the lockdowns in healthy urban populations. The city context is particularly interesting because infectious diseases are known to spread more easily in densely populated urban areas [4]. At the same time, there is less space available per capita in urban compared to rural areas, hence avoiding public places and crowds during a lockdown leads to significant changes in the typical sensory experience of a city dweller (for instance, more time spent indoors, more screen time, less time seeing other people). This, in turn, may have led to modulations in the visual and emotional processing of the urban environment.

Functional near-infrared spectroscopy (fNIRS) is a relatively recent brain-imaging technique providing a non-invasive and robust measurement of the light intensity changes (wavelengths between 650 nm and 1000 nm) caused by the concentration of oxygenated hemoglobin (oxy-Hb) and deoxygenated hemoglobin (deoxy-Hb) in the brain vessels [5]. The brain’s neural activity in certain regions triggers an increase in blood flow and volume in those regions, which is disproportionately higher than the metabolic demand [6]. Even passive exposure to urban scenes requires cognitive resources to process sensory stimuli, which can be reflected as a brain activity pattern on the fNIRS scan. These patterns occur in certain regions of the brain cortex. For the scope of this study, the frontal and occipital regions are particularly important to focus on. The frontal region (or frontal lobe) is one of the most important neuroanatomical structures and plays an important role in emotional [7] and cognitive [8] task processing, among other executive functions. The occipital lobe, on the other hand, is commonly associated with visual processing and spatial orientation [9,10].

Previous brain imaging research shows that the level of cognitive effort or attention pattern utilized to process different scenes can vary. More specifically, it has been shown that scenes of nature induce less strenuous, involuntary attention, [11,12] and less activity associated with stress and anxiety [13,14] when compared to busy urban scenes (with noise, cars, buildings, and crowds). These findings support the Attention Restoration Theory (ART), according to which an exposure to nature has a soothing, balancing effect on emotions and can restore depleted attention, while also reducing mental fatigue [15,16,17]. Nevertheless, the brain hemodynamic patterns underlying exposure to urban and natural scenes in the context of a COVID-19-related lockdown remains unknown. Our parallel electroencephalography (EEG) study with a similar design confirmed the general decline in mood and approach-related motivation towards urban scenes after the lockdown when compared to before [18]. However, visual attention patterns were not tested in our EEG study.

To fill this gap in knowledge, the purpose of the current experiment is to explore the impact of the lockdown with stay-at-home orders on the hemodynamic, frontal, and occipital response of a healthy adult population at three urban sites in the highly urbanized country of Singapore, in a controlled, laboratory-based longitudinal study. We expect that the considerable change of lifestyle imposed by the lockdown contributed to general changes in the brain’s hemodynamic patterns, which will manifest as modulations in the frontal and occipital regions of the brain. Given the previous findings in environmental neuroscience, we also expect to observe varying brain reactivity towards different types of urban spaces, with green spaces generating a more soothing effect on the brain as compared to urban scenes.

## 2. Materials and Methods

### 2.1. Participants

We recruited 25 right-handed, healthy subjects (11 male, M_age_ = 40.73, SD_age_ = 18.27, range 21 to 74, and 14 female, M_age_ = 39.50, SD_age_= 17,94, range 29 to 69) for two fNIRS scans based in the same laboratory room at the university premises. The first scan (T1) took place between the second quarter of 2019 and the first quarter of 2020. The last participant for T1 was accepted for a scan four days before the first COVID-19-related restrictions affecting public space were announced by the Singapore Ministry of Health (introducing body temperature screening at the entrance to selected public spaces) on 20 January 2020 [19]. The second scan (T2) took place between 2 and 30 June 2020—right after the 56-day-long lockdown in Singapore was over. None of the participants had any previous history of psychiatric, neurological, or chronic medical diseases. The study protocol was approved by the National University of Singapore’s Ethics Committee (NUS-IRB_S-20-12).

### 2.2. Study Protocol and Setup

After signing the informed consent, participants had a portable fNIRS cap set on their head, followed by the light-blocking over-cap. They were then instructed to sit comfortably on the chair and passively watch the presentation of nine fixed-frame videos repeated three times, in a random order, displayed on a 108 × 178 cm roll-up screen positioned about 200 cm in front of their eyes. There was a 60 s long resting state preceding the video presentations, where participants were looking at a gray, empty screen. Videos were 20 s long (passive task) with a 15 s pause in between, at which time a fixation cross was displayed on the screen (Figure 1). The videos were projected using the HD29 Darbee Optoma Home Theatre Full HD projector with 1080p (1920 × 1080) screen resolution. The natural sound, as recorded, was also played with the video using standard PC audio speakers placed near the projector, behind the participant’s chair. During the experiment, a daylight-imitating lamp (with a light hue of 5500 K) was turned on.

### 2.3. Stimuli

Presentation of stimuli consisted of nine scenes captured at three public urban spaces in Singapore (Figure 2). Two of these locations were green spaces where no people were within view: *Urban Park*, where scenes included lush greenery settings, and the *Neighborhood Landscape,* which besides vegetation included more paved areas and residential buildings than the park. The third location, *City Center,* had negligible natural elements, plentiful views of buildings, infrastructure, people (without masks, as videos were filmed before the pandemic), and street traffic. All videos were recorded with a Sony™ HDR-TD30 camcorder with a frame size of 1920 × 1080 pixels, 50 frames per second, and 16:9 aspect ratio in TS(AVC)(“*.m2ts”) format, and were 20 s long. The video files were converted to MP4 format supported by presentation software (compression rate 50:1), using VideoProc 3.2 (Digiarty Software Inc.). All videos were comparable in terms of luminance (M = 124.0, SD = 7.1) and root-mean-square contrast (M = 52.6, SD = 3.4), and were not processed further, so as to avoid low-level influence on attentional processing [20].

### 2.4. Measurement of Brain Activity

Two portable NIRS SPORT devices by NIRx^®^ (NIRx Medical Technologies, LLC, Berlin, Germany), with eight sources and eight detectors each, were combined in tandem mode onto a stretchable aniCAP (supplied by NIRx^®^) according to a pre-set prefrontal-occipital montage (Figure 3). Cortical hemodynamics were measured with two wavelengths of near-infrared light (760 and 850 nm) and the sampling rate was set at 3.47 Hz. Positional data of sources and detectors were obtained for two non-participants of fNIRS measurement (1 male, 1 female, both age 24) using a 3D digitizer supplied by NIRx^®^.

For spatial profiling of fNIRS data, we adopted the probabilistic registration method [21,22,23] to register the data to the Montreal Neurological Institute’s (MNI) (Montreal, QC, Canada) standard brain space. The macro-anatomical labeling was based on [24].

### 2.5. Data Processing and Analysis

Signals reflecting the oxygenated hemoglobin (oxy-Hb) and deoxygenated hemoglobin (deoxy-Hb) changes were calculated in units of millimolar-millimeter (mM-mm) using Homer 3 [25].

For the statistical analysis, the General Linear Model (GLM) method was used, which was optimized for fNIRS data to calculate beta values (response amplitudes) [26].

We analyzed changes in oxy-Hb signal because of its higher sensitivity to changes in cerebral blood flow than that of deoxy-Hb and total-Hb signals [27,28,29], its higher signal-to-noise ratio [28], and its higher retest reliability [30]. First, we preprocessed the individual time series data for the oxy-Hb of each channel. The channels with signal variation of 10% or less were considered defective measurements, and therefore excluded from the analysis. Second, we removed the influence of measurement noise, such as breathing and cardiac movement, from the remaining channels using the Wavelet-Minimum Description Length (Wavelet-MDL) detrending algorithm [31].

After preprocessing, we analyzed in Matlab 2007b with the tools from [26] using the adaptive GLM, and by regressing the data with a linear combination of explanatory variables, i.e., regressors and an error term. We generated the regressors (Equation (2)) by convolving the boxcar function *N*(*τ_p_*, *t*) with the hemodynamic response function (HRF), shown in Equation (1) [32].
(1)$$hh\left(\tau_{p},\;t\right)=\frac{t^{\tau_{p}}e^{-t}}{\left(\tau_{p}\right)!}-\frac{t^{\tau_{p}+t_{d}}e^{-t}}{A\left(\tau_{p}+\tau_{d}\right)!}$$
(2)$$f\left(\tau_{p},t\right)=h\left(\tau_{p},t\right)*N$$

We set the first peak delay, *τ_p_*, at 6 s (which is a typical set in most fMRI studies), the second peak delay, *τ_d_*, at 16 s and A—the amplitude ratio between the first and second peak—at 6. The first and second derivatives were included in order to further remove the influence of noise on individual data. Regressors included were the rest-fixation cross (Rest) and the passive task-video (Stimuli) for each condition. The β value is used as an estimate of the HRF prediction of the oxy-Hb signal. β1, β4 and β7 are coefficients for the HRF of each condition. β2, β5 and β8 are coefficients for first derivatives on the HRF of each condition. β3, β6 and β9 are coefficients for second derivatives on the HRF of each condition. β10, β11 and β12 are coefficients for the HRF and first and second derivatives of Rest. β13 is a constant term. A total of thirteen β values were calculated. An example of design matrix X is shown in Figure 4.

For each channel and each condition, the β-values obtained were subjected to group analyses. First, we screened for the channels that exhibited any possible lockdown-related change between the first and second scans (T1 and T2). Specifically, we calculated the contrasts between T1 and T2, and performed one-sample *t*-tests (two-tails) against zero. We set the statistical threshold at 0.05, with the Meff (Effective Multiplicity Correction Method) correction used for family-wise errors [33]. Next, we further analyzed the effects of the time of scans (before and after the lockdown, Table 1 and Table 2, respectively) and sites (S1, S2, S3), with 2 × 3 repeated-measures analysis of variance (ANOVA) on the channels that exhibited possible hemodynamic changes according to the one-sample *t*-test described above. We corrected the *p* value by Bonferroni method for family-wise errors. For reporting statistics, we used the 6th edition of the American Psychological Association’s (APA) statistic standards [34], where *p* < 0.05 is considered significant, and a *p* value between 0.05 and 0.1 can be considered marginally significant, with the latter especially prevalent in the area of experimental psychology [35].

## 3. Results

Due to the data loss, caused by poor data quality due to thick hair and the contamination of the fNIRS signals by task-evoked physiological noise, only data from 12 out of the initial 25 participants (5 male, M_age_ = 47.80 SD_age_ = 17.79, range 25 to 74, and 7 female, M_age_ = 39.00, SD_age_ = 18.18, range 21 to 65) were included. They were all Singapore residents with normal or corrected-to-normal vision; seven were Chinese, four were Indian, and one was of another ethnicity. They were mainly university-educated (*n* = 10), living in a five-person (four bedroom) household (*n* = 5), four-person (three bedroom) household (*n* = 3), or three-person (two bedroom) household (*n* = 3). One participant declared living alone in a one-bedroom apartment.

### fNIRS Results

We analyzed the contrast between T1 and T2 to investigate any effects of the lockdown on oxy-Hb changes during the presentation of each scene. For site 1, we found a significant difference in the CH9 (one-sample *t*-test, *p* < 0.05, Cohen’s d = −1.01, Table 1 and Table 2) and a marginally significant difference in the CH19, but a moderate effect size was obtained (one-sample *t*-test, *p* > 0.05, Cohen’s d = −0.92, Table 1 and Table 2) with a Meff value of 9.41. For site 2, we found a marginally significant difference, but a moderate effect size was obtained in the CH9 (one-sample *t*-test, *p* > 0.05, Cohen’s d = −1.00, Table 1 and Table 2) and the CH19 (one-sample *t*-test, *p* > 0.05, Cohen’s d = −0.93, Table 1 and Table 2) with a Meff value of 9.22. For site 3, we found a marginally significant difference, but the moderate effect size was obtained in the CH10 (one-sample *t*-test, *p* > 0.05, Cohen’s d = −0.94, Table 1 and Table 2) and the CH27 (one-sample *t*-test, *p* > 0.05, Cohen’s d = −0.91, Table 1) with a Meff value of 9.34.

According to the initial screening results, we conducted the 2 × 3 repeated-measured ANOVA on CH9, CH10, CH19 and CH27. For CH9, the result showed a significant difference in the scan time factor (F(1,11) = 12.546, *p* = 0.019, η_p_^2^ = 0.533). We found no significant differences in the site factor (F(2,22) = 2.624, *p* = 0.380, η_p_^2^ = 0.193) or in the interaction effects (F(2,22) = 0.7329, *p* = 1.968, η_p_^2^ = 0.062). For CH10, we found no significant differences in the scan time factor (F(1,11) = 6.242, *p* = 0.118, η_p_^2^ = 0.362), the site factor (F(2,22) = 0.0396, *p* = 3.845, η_p_^2^ = 0.004), or the interaction effects (F(2,22) = 1.222, *p* = 1.256, η_p_^2^ = 0.100). For CH19, the result showed a significant difference in the scan time factor (F(1,11) = 11.511, *p* = 0.024, η_p_^2^ = 0.511). Still, we found no significant differences in the site factor (F(2,22) = 0.703, *p* = 2.024, η_p_^2^ = 0.060) or the interaction effects (F(2,22) = 0.4978, *p* = 2.458, η_p_^2^ = 0.043). For CH27, we found no significant differences in the scan time factor (F(1,11) = 6.670, *p* = 0.102, η_p_^2^ = 0.378), the site factor (F(2,22) = 2.191, *p* = 0.542, η_p_^2^ = 0.166), or the interaction effects (F(2,22) = 0.999, *p* = 1.538, η_p_^2^ = 0.083). Accordingly, we estimated macro-anatomical labels (Table 1) using Brodmann’s atlas [24].

## 4. Discussion

The objective of this study was to explore the impact of the COVID-19-related lockdown on the brain hemodynamics during a passive task exposure to three urban scenes in Singapore: two urban green spaces (*Urban Park* and *Neighborhood Landscape*), and one busy urban core (*City Center*) with negligible greenery. We compared levels of blood oxidation (oxy-Hb) in the frontal and occipital cortex of the same group of 12 individuals before and after the lockdown.

The results indicated the differences in brain hemodynamics based on the different content of the videos. Notably, videos of urban green spaces (*Urban Park* and *Neighborhood Landscape*) triggered a decrease in brain hemodynamics over time in symmetrically positioned channels: CH9 on the left, and CH19 on the right occipital lobe (with CH19 showing marginal statistical significance). At the same time, in the *City Center* condition, we observed a marginally significant decrease in occipital hemodynamics over time, but in the midline occipital region CH10 (Table 2). The difference in symmetry in urban versus green views can be explained by the “contralateral effect”, according to which when visual attention is directed to the right hemifield, attentional modulations are found in the left occipital lobe, and, conversely, visual attention to the left hemifield causes decreased activity in the right visual cortex [9,10]. Visual attention being evenly distributed around the hemifield while observing green spaces suggests *globa**l* visual processing—seeing the space as a whole,—while, in the case of the *City Center*, attention could be *loca**l*—fragmented and focused on processing multiple scattered elements (people, cars, buildings, signage, etc.) [36]. This difference in processing urban and green scenes was demonstrated in previous EEG studies [37] and is related to the restorative and balancing effect that nature scenes have on the mind, according to the Attention Restoration Theory (ART) [15,17].

In our experiment, the *City Center* was the only condition triggering a marginally significant change in the frontal cortex over time. After the lockdown, participants viewing the busy urban scene had significantly lower levels of oxy-Hb in CH27 positioned on the left superior frontal cortex. At the same time, no significant changes in the frontal lobes between T1 and T2 were detected in the green space conditions. This effect coupled with the local fragmented visual attention may suggest more psychological strain while exposed to busy urban scenes as compared to the green space scenes. The potential mechanism of this may be related to aversive attitudes towards busy, crowded public places that emerged during the pandemic, especially since pedestrians present on the *City Center* videos were not wearing the face masks—a previous study has highlighted the association between not wearing masks and increased symptoms of post-traumatic stress disorder during the COVID-19 pandemic [38]. Additionally, videos with urban nature did not include any people or reminders of the pandemic, which could have had a calming effect on participants.

Furthermore, the ANOVA results show that the CH9 and CH19 activation was significantly different between scan times (T1–T2), and that this difference is independent of the site factor. This suggests that the effect of the lockdown on the brain hemodynamics was not region-specific, but rather it holistically affected the brain’s reactivity to all sceneries. We observed that the levels of oxy-Hb in these channels decreased over time (Table 2). Decreased brain hemodynamics are commonly associated with depressive symptoms [39] and aging of the brain [40,41]. The biological mechanisms of these phenomena are not fully understood, but some studies have suggested that depression is linked with the neurons and glial cells losing some of their functions, thereby causing lower levels of regional brain electrical activity, which in turn lead to decreased levels of oxy-Hb [42]. On an emotional level, depression is often characterized by emotional withdrawal or an aversive attitude towards the external stimuli [43]. Aging research, on the other hand, shows that with years passing by, the neural volume and activity tends to decrease [44], which may in turn be linked to a decrease in brain oxy-Hb in certain brain regions. The depressive symptoms occurring post-lockdown seem to be supported by previous behavioral studies [1,2] as well as longitudinal EEG experiments [18]; people experiencing distress, uncertainty, social isolation and stimuli deprivation may have developed depressive symptoms over the lockdown period. It is also plausible that some slower neural reactivity to visual stimuli (typical in older age participants) may have occurred as a result of a prolonged change of everyday routine and limited sensory stimulations. In this regard, studies have highlighted an alarming increase in the use of digital devices during the pandemic among youths and adults [45,46]. Simultaneously, there has been a decrease in physical activity and time spent outdoors [47,48], which are considered as opportunities to relax eye strain by looking at more distant natural objects [49,50]. This may be an important explanation for the general changes in visual processing observed in our study. However, more research is needed to confirm this with evidence, as data regarding participants’ screen time and outdoor activity before and after the lockdown were not collected in this study.

There were several limitations to this study. First, our final sample size was small due to difficulties with recruitment during the early stages of reopening and fears of COVID-19 infection among participants. This was also contributed to by significant data loss due to signal contamination. Nevertheless, our sample size was comparable with other similar fNIRS studies [8,51,52]. Secondly, due to device specificity, we focused only on the frontal and occipital cortices, and were not able to assess deeper neuroanatomical structures [39]. Moreover, we did not control for participants’ individual emotional relationship with the viewed spaces, such as level of familiarity or life events occurring in these spaces between T1 and T2, which may have potentially influenced the brain’s response.

## 5. Conclusions

This study explored the impact of the COVID-19-related lockdown on changes in brain hemodynamics in response to three urban scenes. We found significant changes in the brain’s reactivity to the visual stimuli, potentially linked to the lower mood and more depressive symptoms. Moreover, reduced cognitive functions developed over the lockdown can be attributed to a considerable change in daily routines during prolonged stay-at-home orders (for example, limited physical activity and increased screen time). These symptoms may lead to a heightened risk of mood and cognitive dysfunctions in confined populations. Furthermore, our study highlights the risk of increased psychological strain post-COVID-19 during exposure to typical busy urban street views, and the contrasting importance of green urban space exposure, which could counteract the negative effect of that strain, while also having further beneficial impact on the brain.

## Figures and Tables

**Figure 1 ijerph-18-06118-f001:**
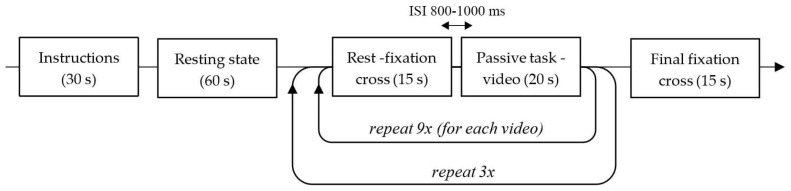
T1 and T2 brain imaging protocol.

**Figure 2 ijerph-18-06118-f002:**
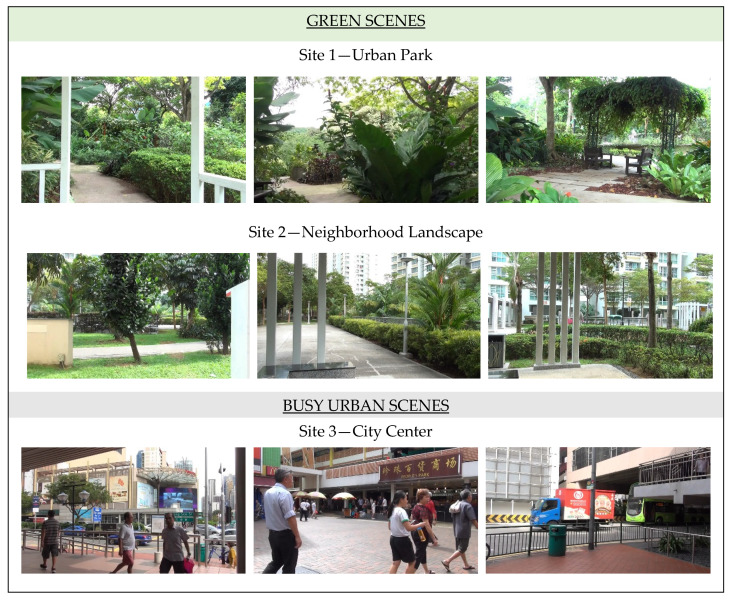
Photo snippets of the still-frame videos presented to participants.

**Figure 3 ijerph-18-06118-f003:**
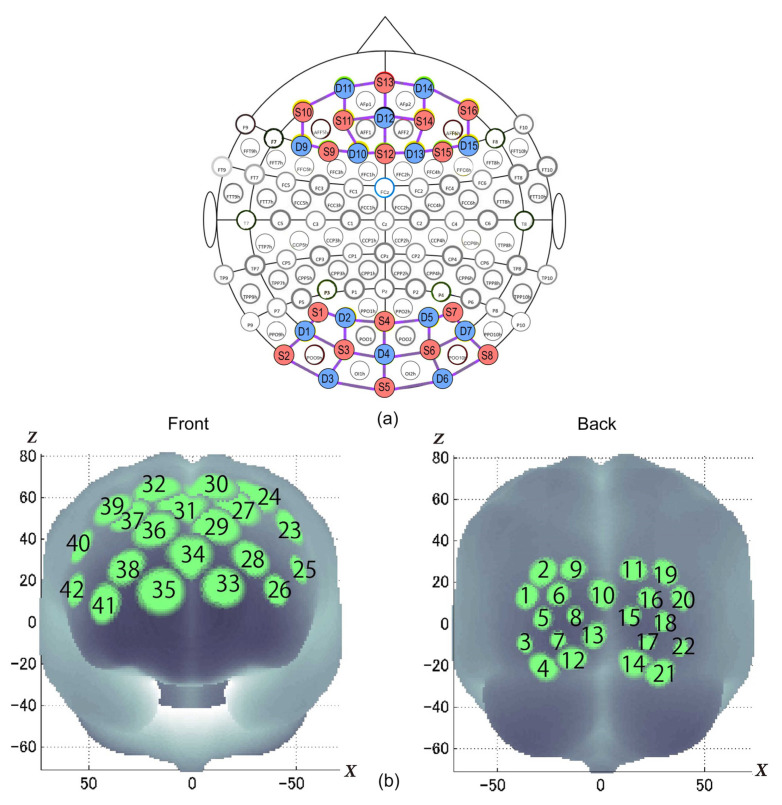
(**a**) Montage map with optodes placement (sources (S) in red and detectors (D) in blue) and (**b**) statistically estimated fNIRS channel locations on the brain for two students, their spatial variability (SDs radii of the light green circles) associated with the estimation exhibited in MNI space. Front and back head views.

**Figure 4 ijerph-18-06118-f004:**
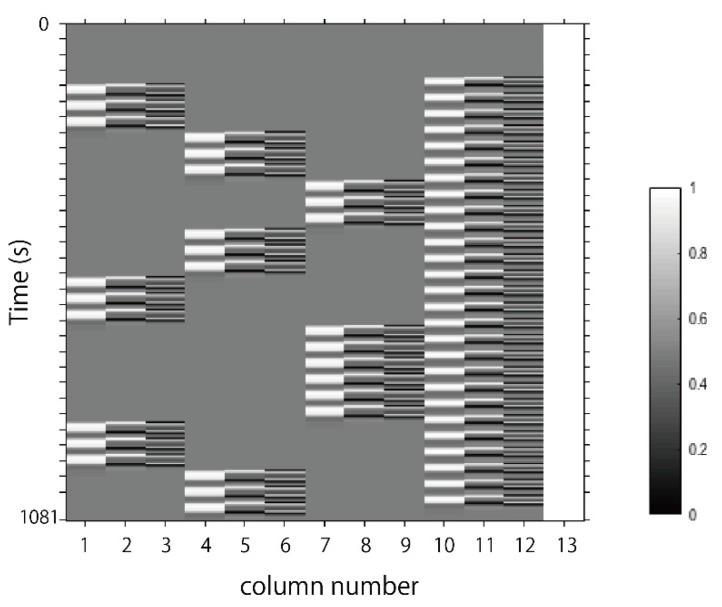
An example of design matrix X. The first peak delay was set as *τ_p_* = 6 s and “Time” indicates the row number of samples. The first to third columns indicate the canonical HRF *f*(*τ_p_*, *t*), the derivatives, and the second derivatives, respectively, for site 1. The fourth to sixth columns indicate the canonical HRF *f*(*τ_p_*, *t*), the derivatives, and the second derivatives, respectively, for site 2. The seventh to ninth columns indicate the canonical HRF *f*(*τ_p_*, *t*), the derivatives, and the second derivatives, respectively, for site 3. The tenth to twelfth columns indicate the canonical HRF *f*(*τ_p_*, *t*), the derivatives, and the second derivatives, respectively, for Rest. The thirteenth column indicates the constant.

**Table 1 ijerph-18-06118-t001:** The MNI coordinates of channel 9, 10, 19, 27 and the probability of the recorded data originating from the listed anatomical locations.

Channel	x	y	z	SD	Anatomy	MAL(%)
9	−13.3	−100.0	26.3	7.2	BA18-Visual Association Cortex (V2), left	74.8
					BA17-Primary Visual Cortex (V1), left	25.2
10	0.3	−101.3	14.3	8.6	BA17-Primary Visual Cortex (V1), left	84.8
					BA18-Visual Association Cortex (V2), left	15.2
19	30.7	−94.0	24.7	7.7	BA18-Visual Association Cortex (V2), right	74.7
					BA19-Visual Association Cortex (V3), right	25.3
27	−21.3	36.3	55.3	13.6	BA8-Frontal eye fields, left	58.1
					BA9-Dorsolateral prefrontal cortex, left	41.9

SD indicates standard deviation in the spatial estimate. BA stands for Brodmann Area. MAL stands for macro-anatomical probability. MAL is based on Roden and Brett (2000).

**Table 2 ijerph-18-06118-t002:** Channels showing marginally significant and significant difference in oxy-Hb signals in contrast with baseline (“T2-T1”).

	Channel	Mean (mM·mm)	SD	*t*	*p*	Sig
Site 1 *Urban Park*	9	−0.225	0.223	−3.50	0.0471	*
	19	−0.164	0.178	−3.19	0.0814	†
Site 2 *Neighborhood Landscape*	9	−0.286	0.287	−3.44	0.0503	†
	19	−0.207	0.223	−3.22	0.0757	†
Site 3 *City Center*	10	−0.365	0.388	−3.26	0.0708	†
	27	−0.207	0.226	−3.16	0.0843	†

Statistical significance is presented as follows: † *p* < 0.1; * *p* < 0.05. Abbreviations: SD, standard deviation; *t*, *t* value; *p*, *p* value; sig, statistical significance.

## Data Availability

Exclude this statement.

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
