# Peer review of "Hemodynamic Response to Three Types of Urban Spaces before and after Lockdown during the COVID-19 Pandemic"

_ijerph, 2021, doi:10.3390/ijerph18116118_

Round 1

Reviewer 1 Report

To explore the impact of the COVID-19 related lockdown on the brain hemodynamics, this study investigated the hemodynamic activation of the prefrontal and occipital cortices of 12 participants before the COVID-19 and  after the lockdown in Singapore. The subjects were exposed to 2 green space scenes: Urban Park and Neighbourhood Landscape, and 1 urban space scene: City Centre in Singapore. It was found that a significant decrease in hemodynamic activation after the lockdown and that the green spaces stimuli decreased hemodynamic activation in the left Visual Association Cortex  (CH9). The authors suggest that “e COVID-19-related lockdown experienced by urban inhabitants contributed to decreased brain hemodynamics related to heightened risk of mental health disorders such as depression or cognitive functions decline.”

Major concerns include:

  • Inconsistent data reporting. It was mentioned that “7 female, Mage=42.67, SDage=18.54”. However, in Materials and Methods, it was mentioned “14 female, Mage=40.04, SDage=17.94” and in Discussion, it was mentioned “f 12 individuals”. Provide the means and SDs for female and male, and age ranges in the Materials and Methods.
  • Inappropriate significance reporting. There is only significant or not significant. Please exclude the conclusions from the findings p<0.05 from the entire text.  There is no such thing called “marginally significant” if p<0.05. P<0.05 is not significant.
  • Conclusions out of scope. The authors' conclusions are out of scope of this study. For example, how “plausible” to conclude that “the COVID-19-related lockdown experienced by urban inhabitants contributed to decreased brain hemodynamics related to heightened risk of mental health disorders such as depression or cognitive functions decline” based on the finding that “a significant decrease in hemodynamic activation after the lockdown and that the green spaces stimuli decreased hemodynamic activation in the left Visual Association Cortex  (CH9)” ? There is no  measurement to access the "risk of mental health disorders" of the subjects included in this study.

Other specific concerns: 

Abstract:

  • “We observed a significant or marginally significant decrease in average Oxy-Hb over time for each of the visual conditions” There is only significant or not significant. It is not clear what “marginally significant” means.
  • Please also provide the names of the ROIs of all the channels covering. The numbers of the channel meant nothing if readers are not reading the main text.
  • Spell out “Oxy-Hb”.
  • Name Singapore.

Materials and Methods

  • Figure 1: label each block that were blank in the current version and the time durations. It will be reader-friendly in terms of the study design.
  • Figure 3: please make the sizes of the two sub-figures at the bottom panel the same as the one on top. Each sub-figure should be in the same size for better visualization.

Results

  • Include the number of subjects included in the study for data analyses. Provide the means,  SDs and the age ranges for female and male, accordingly.
  • Remove Figure 5. Basically, Figure 5 refers to Table 2. The information provided in Table 2 is sufficient enough.

Discussion

  • “The goal of this study was to…” should be changed to “the objective of this study..”
  • “Decreased brain hemodynamics is commonly associated with depressive symptoms [33] and ageing of the brain” It needs grammatic correction.

Author Response

Answers to Reviewer are provided in the attached pdf file. 

Reviewer 2 Report

This is a nice companion study to the author's EEG publication. It is certainly worth publishing and nicely written/conducted. I am not familiar with fNIRs, however, and strongly suggest an expert reviewer in this technique read the methods and results too.

A definition of hemodynamics in the introduction would be nice. I actually had no further comments. Well done (from my naive-fNIRS perspective)!

Author Response

Thank you very much for your positive feedback about our work.

Reviewer 3 Report

    1)    The abstract is supposed to follow the structure background/methods/results/conclusions, but it’s not suppose to have those titles included.

    2)    Do not start a sentence with a number. 25 right-handed, healthy subjects were recruited… can be replaced by We recruited 25 right-handed, healthy subjects…

    3)    The participants were shown scenes of locations probably familiar to them. If I was the participants and I had a recent prior negative experience in the location shown the data would probably be skewed by that. Did you question the participants about how familiar they are with the scenes shown?

    4)    Did you consider showing scenes from locations they are less likely to know personally? That would assure that the results wouldn’t be influenced by the participants’ recent personal experiences in those locations. 

    5)    You started your first scans T1 in the second quarter of 2019, that means what, April-June 2019? So that was pre-pandemic? So, some of the T1 scans were done before the pandemic and some were done during the pandemic (first quarter of 2020). That means you can’t really compare T1 with T2. During January - March 2020 people already knew about the virus spreading around even if lockdowns weren’t implemented yet. But knowing about the virus (even without lockdown) would already skew the results.

    6)    To save the paper you should also compare T1 you made before the virus started spreading in Wuhan with T2 (corresponding participants); and then make a separate comparison of T1 scans you made after the virus was known with T2 (corresponding participants). Otherwise we would never know if the differences are because of seeing places with the virus spreading around or because of seeing open spaces after experiencing lockdowns.

Author Response

Authors reponse to the Reviewer's comments can be found in the attached pdf file

Round 2

Reviewer 1 Report

The quality of the manuscript has been improved significantly by the authors. However, there are points to be addressed before publication.

  1. The authors addressed the inconsistent sample size in the authors response letter and revised in the manuscript accordingly. Please also state the reason of data lose in the manuscript.
  2. P>0.05 is generally considered not significant. If the authors follow the APA statistic standards, please explicitly explain the definition, with citation, and why it was chosen for statistical analysis and data reporting in the Method session. I am sorry I made a wrong symbol in previous letter. Basically, “marginally significant” are not used normally and p>0.05 are not considered significant.
  3. Figure 1: Please add the duration of ISI in the figure.

Author Response

We thank the Reviewer for the answers and further comments. We have addressed them all in the updated version of the manuscript, marked in blue. Please see detailed answers below: 

1) We have added the cause of data loss, which was poor data quality due to thick hair and the contamination of the fNIRS signals by task-evoked physiological noise (line 219)

2) We have added the  APA standards description to support reporting of the marginally significant results (lines 212-216).

3) The duration of ISI was addedd in figure 1. 

Reviewer 3 Report

You disagreed that knowing about the pandemic would skew the results of your study because you focused on the lockdown. But you didn't provide an explanation why you disagreed. Some T1 scans are pre-pandemic, and other T1 scans are after the virus was known spreading. That alone can already make difference in those two groups of T1 scans alone. You should at the very least discuss that in your paper.

Author Response

We thank the Reviewer for another round of revision.

We would like to again, perhaps more thoroughly explain why we think we should not address the Reviewer's query as a limitation to our study because media release-related stress wasn't the scope of our study. We worry this may confuse the readers.

What was the scope of our study was the lockdown-related stress response, so technically, any time before the lockdown (even one day before the lockdown, when compared to right after lockdown), would be a correct approach to test the lockdown effect. 

Nevertheless, we agree with the Reviewer that the selection of recruitment dates is important and could be better justified in the Methods section. This is why we have modified some methodology reporting to include more details about dates of recordings, and worldwide events at that time. 

(line 104) "The first scan (T1) took place between the second quarter of 2019 and the first quarter of 2020 (the last participant for T1 was accepted for a scan four days before the first COVID-19 related restrictions affecting public space use were announced by the Singapore Ministry of Health [ref], introducing body temperature screening at the entrance to selected public spaces - on January 20th, 2020). The second scan (T2) took place between 2nd and 30th June 2020 – right after the 56-day-long lockdown in Singapore was over." 

The last two participants in T1 had their scans on 15.01.2020 and 16.01.2020, and only those 2 cases would be presumably of the Reviewer's concern. (The previous scan were taken on 1st of October 2019 and before). 

Around that time (15-16 January) in Singapore, indeed we heard the media release about the virus spreading in China, but no measurements were taken by the government. Life in the urban space still looked pretty normal.

Our research team decided to make the cutoff inclusion date of participants to be the first date when something significantly changed in the participants' life, and in the public space (this study is about urban space). We believe this was a correct approach, that allowed us to test our hypothesis with the scope of looking for differences before and after the lockdown.